# Perception, regulation, and effects on longevity of pollen fatty acids in the honey bee, *Apis mellifera*

Marielle C. Schleifer[1,2], Fabian A. Ruedenauer[2], Johanna Ziegler[1], Sara D. Leonhardt[2‡], Johannes Spaethe[1‡]*

**1** Department of Behavioral Physiology and Sociobiology, Biocenter, University of Würzburg, Würzburg, Germany, **2** Plant-Insect Interactions, Department of Life Science Systems, Technical University of Munich, Freising, Germany

‡ SDL and JS share senior authorship on this work.
* johannes.spaethe@uni-wuerzburg.de

**Data Availability Statement:** All relevant data is available on https://doi.org/10.17605/osf.io/geuzv

## Abstract

For successful cross-pollination, most flowering plants rely on insects as pollinators and attract them by offering rewards, predominantly nectar and pollen. Bees–a highly important pollinator group—are especially dependent on pollen as their main source of essential nutrients, including proteins, lipids, and sterols. Fatty acids (FAs) in particular play a pivotal role as fundamental energy source, contributing to membrane structure integrity, cellular homeostasis, and cognitive processes. However, overconsumption of FAs can have detrimental effects on fitness and survival. Thus, bees need to precisely modulate FA intake. To better understand how *Apis mellifera*, the globally predominant managed pollinator, regulate FA intake, we conducted controlled feeding experiments with newly hatched honey bee workers by providing pollen with different FA concentrations. We additionally investigated the honey bee's capacity to perceive individual FAs by means of chemotactile proboscis extension response (PER) conditioning. We tested both natural concentrations and concentrations exceeding those typically found in pollen. Given the dose-dependent importance of FAs observed in other bee species, we hypothesized that (i) a high FA concentration in pollen would reduce honey bee longevity, and (ii) honey bees are able to perceive individual FAs and differentiate between different FA concentrations via antennal sensation prior to consumption. Our study revealed that elevated FA concentrations in pollen resulted in reduced consumption rates and increased mortality in *Apis mellifera*. Workers can detect and discriminate between saturated and unsaturated FAs utilizing their antennae. Moreover, they were able to distinguish between individual FAs and also between different concentrations of the same FAs. Our results suggest a high sensitivity of *A. mellifera* towards both the concentration and composition of individual FAs, which greatly impacts their foraging decisions and fitness. These insights contribute to the growing evidence highlighting the importance of balanced nutrient ratios, in particular of FAs, for bees and other organisms.

**Funding:** Deutsche Forschungsgemeinschaft (DFG project: LE 2750/5-2 to SDL and SP1380/1-2 to JS).

**Competing interests:** The authors have declared that no competing interests exist.

## Introduction

All animals, including humans, need to balance their intake of macronutrients to optimize their health and fitness [1]. However, not all resources provide an adequate nutrient balance due to variation in nutritional compositions. Animals consuming non-complementary, nutritionally imbalanced diets must therefore reach a suitable compromise between overconsumption of certain nutrients and underconsumption of others [2, 3].

Florivores, like bees, rely exclusively on flowering plants as their food source. While adult bees primarily consume nectar as their source of energy (carbohydrates), they also collect pollen, especially for feeding their larvae [4–6]. Pollen provides them with all other needed nutrients, i.e. proteins (in the form of amino acids (AA)), lipids (comprising fatty acids (FA) and sterols), minerals, vitamins and various plant secondary metabolites (PSMs, e.g. nicotine or caffeine) [7]. However, pollen nutrient content (often referred to as nutritional quality) can significantly vary within and among different plant species [8–12]. For example, protein to lipid (P:L) ratio in honey bee collected pollen can range from 1.3 to 13.1 [13–15]. Differences in pollen nutrient composition have been demonstrated to strongly impact bee physiology [16, 17]. Especially under challenging environmental conditions and when facing stressors, such as insecticides, low P:L ratios appear to support bee health [13]. These findings underscore the importance of nutritionally appropriate diets for bee health and fitness and point to a prominent role of lipids, such as FAs.

There are four distinct categories of FAs: i) saturated FA (SFAs), ii) mono-unsaturated FAs (MUFAs) or omega-9 poly-unsaturated FAs (PUFAs), iii) omega-6 PUFAs and iv) omega-3 PUFAs. Among these, only two types are essential nutrients for many animals: omega-6 PUFAs (e.g., linoleic acid, primarily found in seeds) and omega-3 PUFAs (e.g., linolenic acid, primarily found in green leaves). They are essential, because they cannot be synthesized by animals and cannot be converted into one another [18]. Honey bee physiology (i.e., brood development, adult longevity and body FA composition) is notably affected by dietary lipid concentrations and omega-6:3 ratios: Comparatively higher concentrations of 8% of total dietary lipids were found to increase brood production, whereas high omega-6:3 ratios increased mortality and diminished brood rearing [19]. Similarly, in bumble bees, diets high in FAs also decreased reproductive fitness and survival [20]. Moreover, a balanced omega-6:3 FA ratio supported the cognitive capabilities of young nurse bees [21, 22], whereas an omega-3 deficiency was linked to impaired learning [23]. Additionally, omega-3 deficiency resulted in smaller hypopharyngeal glands, which are responsible for the production of royal jelly [23]. On top, diets high in FAs were found to modulate the composition of gut microbiota in honey bees and cause overweight [24]. Beyond their nutritional and physiological roles, some FAs possess antimicrobial properties that can significantly impact honey bee health. In particular, FAs with up to 14 carbon atoms and/or with one or more double bounds show antimicrobial activity [25], which can protect bees against pathogenic microorganisms.

Consequently, optimal bee development, health and cognitive functioning appear to require a minimum absolute quantity of essential FAs alongside a balanced omega-6:3 ratio [26]. Overall, omega-6 PUFAs and omega-3 PUFAs are only needed in very small quantities [18]. Their main role is as constitutes of membrane lipids, with species-specific membrane FA compositions [18]. In fact, the omega-6:3 ratios observed in pollen mixtures collected by bees show less variability (ranging from 0.3 to 0.9) compared to hand-collected pollen sourced from 28 distinct plant species, where ratios ranged from 0.09 to 5.34 [23], which indicates that honey bees regulate both the quantity and ratio of FA intake.

Such a nutrient-sensitive regulation can only be achieved when the bees are able to perceive specific nutrients. Indeed, both honey bees and bumble bees were found to discriminate

between pollen differing in nutrient content by means of their antennae prior to consumption [27, 28]. Utilizing their antennae as sensory organs, these bees employ chemotactile sensation, also referred to as taste, to evaluate pollen quality prior to ingestion [27]. This pre-ingestive assessment is crucial, given that most bee species, with the exception of a few *Hylaeus* spp. [29], transport collected pollen for larval provisioning externally on their bodies, e.g. on their corbiculae or scopae [30], rather than consuming it themselves. Ruedenauer [20] additionally showed that bumble bees can differentiate between pollen enriched with single FAs, indicating a fine-tuned perception for FAs. Interestingly, while bumble bees could detect single FAs, they were unable to distinguish between pollen samples with varying AA concentrations [20]. Likewise, bumble bees could perceive sterols when presented in isolation (single sterol solved in chloroform), but not when presented in varying concentrations in pollen [31]. These findings suggest that FAs play a particularly important role for bees, with direct effects on their foraging behavior, development, survival and cognition.

It remains unclear whether honey bees are also able to perceive different FAs–a prerequisite for pre-ingestive assessment of FA profiles and hence nutrient regulation. Given the diverse nutrient composition of pollen, even within plant species [7, 10], it is unlikely for bees to regulate all pollen nutrients comprehensively, considering the substantial resources such as time and metabolic energy required for reception (e.g., receptors), perception (e.g. neuronal circuits), and behavioral responses (e.g., learning and memory) [32]. Therefore, we postulate that only those nutrients or compound groups are perceived and regulated by bees, that are detrimental to their health and fitness when over- and/or underconsumed, whereas compounds that are not maleficent are not perceived or at least may receive less attention from bees [32].

In this study, we investigated whether honey bees regulate the intake of diets varying in FA content and how high amounts of FAs affected the bees' longevity by conducting no-choice feeding assays. Furthermore, we examined whether honey bee workers were able to perceive and discriminate different individual FAs (presented in isolation within a solvent) as well as different concentrations of FA mixtures using chemotactile proboscis extension response (PER) conditioning. Based on the known importance of FAs and on findings in bumble bees [20, 28], we hypothesized that (i) honey bees are able to perceive and discriminate different FAs via chemotactile sensation of their antennae (prior to consumption), and that (ii) variation in FA concentrations in pollen affects the bees' longevity, e.g., higher FA concentrations have a detrimental effect on bee health.

## Material and methods

### Colonies

The experiments were conducted with workers of the Western honey bee (*Apis mellifera carnica*) kept in Dadant bee hives at the Biocenter of the University of Würzburg, Germany. The surrounding landscape comprised hedges, gardens, grassland, and orchards, which provided the colonies with a diverse range of different flowering plants [33].

### FA analysis

The FA content of the honey bee collected pollen that was used for the feeding experiments was analyzed according a protocol recently established in our lab using gas chromatography-mass spectrometry [34]. For a detailed protocol see **S1 Text**.

## Feeding experiments

The feeding experiments were conducted in June 2022 and lasted for 21 days. A frame of mature brood was taken from a colony and transferred to an emergence box stored in an incubator (33°C, 55% RH, darkness) [19, 21]. Due to a shortage in mature brood frames, we obtained bees from only one colony. On the next day, we collected newly emerged bees (0-24hrs old), since the consumption of pollen by adult bees declines rapidly after the first week [35]. Every bee was exposed to only one out of four different treatments, with each treatment comprising ten replicates. Each replicate consisted of 30 bees placed in a wooden cage (14x8x18cm), resulting in 300 newly emerged bees per treatment. Wooden cages were equipped with wire mesh at the bottom for air ventilation, a removable glass front panel, and two small openings at the top for food provisioning. The caged bees were provided with 50% w/w sucrose solution *ad libitum* and with one of the following four dietary treatments: pure pollen (as a control), low FA–pollen (same pollen enriched with a FA mix resulting in a 1.5 times higher concentration than the control), medium FA–pollen (fivefold increase of FA concentration) and high FA–pollen (tenfold increase of FA concentration). The diets comprised bee-collected pollen (Naturwaren Niederrhein GmbH, Goch-Asperden, Germany), which was ground into a fine powder using a coffee grinder (Bosch, Germany) and mixtures of FAs. The FA mixture comprised several saturated FAs (arachidic, behenic, capric, lauric, myristic, palmitic, and stearic acid), two poly-unsaturated FAs (PUFAs, linolenic (omega-3, essential) and linoleic (omega-6, essential)) and one mono-unsaturated FA (MUFA, oleic (omega-9) acid, for details, see **Table 1**). The pollen–FA- mixture was thoroughly homogenized, and the resulting pollen powder was blended 1:1 with deionized water to achieve uniform texture. Fresh diets were prepared daily and offered to bees on plastic weighing dishes (4cm in diameter). Daily pollen consumption was recorded by weighing the dishes before and after exposure to the colonies and averaged per bee. To account for evaporation-induced weight loss, a control weighing dish containing pollen paste was placed in an empty wooden cage. Sugar water was replaced daily with freshly prepared solution. All procedures were conducted under red light conditions to minimize disturbance. Dead bees were counted and removed without replacement. Bees tested in the feeding experiment were not used for the PER experiments.

## PER experiments

The PER experiments were conducted from April until September in 2022 and 2023 with foragers sourced from ten different unrelated honey bee colonies. On sunny and warm days, five

**Table 1. Fatty acid (FA) concentrations.** Total FA concentrations [mg/g] of honey bee collected pollen (i.e., pure pollen and the same pollen enriched with the 1.5 fold, 5 fold and 10 fold FA concentration as found in pure pollen), percentage of total FAs in pollen and company from which the FAs were obtained.

| FA mg/g pollen | Pollen pure | x1.5 FA | x5 FA | x10 FA | % of total FA content | Source company |
|---|---|---|---|---|---|---|
| Capric | 0.58 | 0.87 | 2.90 | 5.79 | 2% | Carl Roth, Karlsruhe, Germany |
| Lauric | 0.70 | 1.06 | 3.52 | 7.04 | 2% | Sigma-Aldrich, Saint Louis, MO |
| Myristic | 2.68 | 4.02 | 13.40 | 26.79 | 8% | TCI-Europe, Zwijndrecht, Belgium |
| Palmitic | 18.05 | 27.08 | 90.27 | 180.54 | 56% | Sigma-Aldrich, Saint Louis, MO |
| Stearic | 6.09 | 9.14 | 30.46 | 60.92 | 19% | Sigma-Aldrich, Saint Louis, MO |
| Arachidic | 0.55 | 0.82 | 2.74 | 5.49 | 2% | Sigma-Aldrich, Saint Louis, MO |
| Behenic | 0.86 | 1.29 | 4.31 | 8.61 | 3% | Fluka AG (Honeywell), Morristown,NJ |
| Oleic + Linolenic | 1.84 | 2.76 | 9.20 | 18.41 | 6% | Sigma-Aldrich, Saint Louis, MO |
| Linoleic | 0.81 | 1.22 | 4.07 | 8.14 | 3% | Sigma-Aldrich, Saint Louis, MO |
| Total FA content | 32.17 | 48.26 | 160.87 | 321.73 | | |

leaving foragers were randomly collected from each of three hives (15 specimens in total per day) and stored in separate containers. Since the cognitive abilities of honey bees are primarily affected by their social role in the colony rather than their chronological age [36–39], we did not control for the foragers' age. The containers were put on ice for ten minutes to temporally immobilize the bees. The individuals were then transferred into metal tubes and fixed with a 1mm wide adhesive tape placed directly behind the head of the bee. Another tape was wrapped around the tube and the body to prevent further movement. Harnessed like this, the bees could only move their antennae and extend their proboscis. Five minutes after harnessing, the bees were fed with 30% w/w sucrose solution and then starved for three hours in complete darkness. Prior to the experiment, the motivational state of each bee was tested by touching its antenna with a toothpick soaked in 30% w/w sucrose solution. Only bees that showed a proper proboscis extension response (PER) were used for the experiment.

## Experimental procedure

To investigate whether bees can detect and discriminate FAs, we used the differential chemotactile proboscis extension response (PER) conditioning approach [27]. It is based on the principles of classical conditioning [40], which had been adapted for bees by Bitterman et al. [41]. Honey bees (as several other bee species) show an innate response towards sucrose solution, i.e. they extend their proboscis upon contact of their antennae, tarsi, or parts of their mouthparts with a sugar solution (= unconditioned stimulus, US). Through pairing this unconditioned stimulus with a conditioned stimulus (= CS), such as an olfactory, visual, or chemical cue, previously not leading to a PER, the bee can learn to associate the CS with the US. Following repeated training bouts, the bee can learn to elicit a PER solely towards the CS—provided it perceives the CS. To test if the bees can differentiate between two stimuli (e.g., two different FAs), only one stimulus is paired with sucrose solution (= rewarded CS, CS+), while the other remains unrewarded (= unrewarded CS, CS-). If the bee can differentiate between the two stimuli, it will, following a training period, extend its proboscis only to the rewarded stimulus (CS+).

Each day, eight new bees were tested individually and for one stimulus pairing only. The tested bee was placed in a forward-shifted rack for easy access to the antennae. It was then allowed to rest for 15 seconds. Next, the CS was presented using a small filter paper soaked with the stimulus solution (e.g. a specific FA in a solvent, see below). The filter paper was placed on a copper plate and moved towards the bee's right antennae using a micromanipulator. Six seconds after the bee's initial antennal contact with the filter paper, a toothpick was presented to the other antennae for three more seconds. When a rewarded stimulus (CS+) was presented, the toothpick was soaked in sucrose solution and the bee was allowed to lick from the toothpick. When an unrewarded stimulus (CS-) was presented, the toothpick was plain. After a total of nine seconds of stimulus presentation (six seconds CS alone plus three seconds CS and US combined), the filter paper and the toothpick were removed. The bee was then allowed to rest for another 15 seconds. The copper plates were cleaned in 75% ethanol and thoroughly dried before being reused. A new filter paper was placed on the copper plate for each test phase. Each bee was tested in an eight-minute interval (inter-trial interval, ITI) and ten times each for the rewarding and the unrewarding stimulus (resulting in a total of 20 trials). The sequence of stimulus presentation was pseudorandomized to ensure that no stimulus was presented consecutively more than twice. Each stimulus was used as CS+ as well as CS-. Both CS+ and CS- were used as first trial stimulus in separate experimental rounds. The proboscis extension responses were documented using the software "Timing protocol" v2.0 [42]. A response to the stimulus was scored as "1", no response to the stimulus was scored as "0",

and no response to sucrose solution was scored as "NA". Bees failing to respond to the sucrose solution five or more times were excluded from the experiment.

## Stimuli

The stimuli used for the experiment comprised FAs dissolved in chloroform. These solutions were prepared on a weekly basis and stored at -20°C until required. We used three unsaturated FAs with 18 carbon atoms each (linolenic acid (omega-3, essential FA), linoleic acid (omega-6, essential FA) and oleic acid (omega-9)) along with two saturated FAs (stearic acid (also 18 carbon atoms) and capric acid (10 carbon atoms)). We selected these five FA, to test if the bees can differentiate between FAs based on the chain length, saturation state or the number of double bounds. The FAs were dissolved individually in chloroform at a ratio of 1:1000. Furthermore, a mixture of ten FAs in chloroform at the concentration of 1:100, 1:1000, and 1:10000 were prepared (see **Table 1**). The selected concentrations were based on natural FA concentrations found in honey bee collected pollen (see **Table 1**). Ten minutes before the experiment, 5 μL of the respected stimulus (or pure chloroform as a solvent control) was applied to small filter paper snippets (5 x 5 mm). At least 100 snippets were prepared per stimulus to ensure sufficient amounts for testing, as we used a new snippet for every round and bee. Before the filter papers were presented to the bees, the snippets were thoroughly dried (i.e., the chloroform evaporated completely).

## Statistical analysis

Statistical analyses were performed using R version 4.1.2 [43]. Prior to further analysis, normal distribution of all data was assessed using a Shapiro-Wilk test, and homogeneity of variance was assessed using a Levene's test, both included in the R package 'car' [44]. The α-level was set to 5%.

## Feeding experiments

The consumption of pollen per individual per day was calculated as follows: The amount of pollen consumed per cage was divided by the number of bees alive. To evaluate the effect of different concentrations of FAs in pollen consumption (per individual day), we used generalized additive models (GAM). Given the pronounced daily (nonlinear) variability in consumption, we opted for an additive modelling approach that applies smoothers to the data over time. Where necessary, data were log-transformed to ensure normal distribution and homogeneity of variances. To more specifically assess differences in mean consumption between different diets during the critical initial seven-day period (when newly emerged honey bees consumed the highest amount of the provided pollen), we used generalized linear models (GLM) and Tukey Honest Significant Differences (TukeyHSD) as post-hoc test. We finally assessed the quantity of FAs consumed per diet, considering variation in the concentration among pollen diets, also using a GLM and TukeyHSD for post-hoc comparison.

To assess differences in the longevity of honey bees exposed to different diets, we used Kaplan-Meier survival statistics to compare median survival times for each diet individually against all others. To account for multiple testing, we adjusted the α-levels using the Bonferroni correction method. The analyses were conducted using the 'survival' package [45] and the 'KMsurv' package [46].

### PER experiments

First, each stimulus pair was tested against the inverted one using a Wilcoxon rank-sum test with continuity correction to assess whether the type of substance that was rewarded, i.e. used as CS+, influenced the bees' responses (please note that each substance was used as CS+ as well as CS-). If no significant differences were found among the substances, the data were combined and analyzed together. Subsequently, a paired Wilcoxon signed rank test with continuity correction was used to test whether the bees could discriminate between the rewarded and the unrewarded stimulus (i.e., whether there is a difference in the number of responses towards CS+ and CS-). In instances where combining the data was inappropriate, separate analyses were conducted for each stimulus pair (e.g., omega-3 vs chloroform: trials with omega-3 as CS + and chloroform as CS+ were tested separately).

## Results

### Feeding experiments

Freshly emerged honey bee workers showed significantly different consumption patterns between pollen diets varying in FA concentrations (**Fig 1**). Throughout the entire experimental period (21 days), we observed a higher pollen consumption of the control (pure pollen) and low FA diet compared to the medium and high FA diets ($F_{1,3}$ = 30.99, *P = 0.001*, **Fig 1**). After seven days, overall pollen consumption decreased with minor fluctuations (**Fig 1**). Therefore, our analysis focused on the first seven days. We found that during this period, bees consumed significantly more pure pollen and low FA diets compared to medium and high FA diets (*P<0.001*, **Fig 2A and S1 Table**). Bees also consumed significantly less pollen from the high FA diet than from the medium FA diet (*P<0.001*, **Fig 2A and S1 Table**). When analyzing the consumption of total amounts of FAs, bees consumed similar amounts of FAs in the control and low FA diet (*P = 0.179)*, as well as in the medium and high FA diet (*P = 0.999*, **Fig 2B and S1 Table**), but significantly less FAs in the medium and high FA diets compared to the pure and low FA diets (*P<0.001*, **Fig 2B and S1 Table**). For more information regarding the pollen and FA consumption in the second and third week see Supplementary Material **S1, S2 Figs** and **S1 Table**.

The diets also differed significantly in their effect on the longevity of honey bees (**Fig 3**). The highest survival probability was found in the low FA diet, followed by the control diet and the medium FA diet. The lowest survival probability was observed for the high FA diet. Significant differences were found between the low and medium FA diets ($\chi^2$ = 4, *P = 0.05*), pure and high FA diets ($\chi^2$ = 9.9, *P = 0.002*) and the low and high FA diets ($\chi^2$ = 6.1, *P = 0.01*).

### PER experiments

Honey bees were able to perceive most FAs when presented in isolation (FA vs chloroform): omega-3 (linolenic acid), omega-6 (linoleic acid), omega-9 (oleic acid) and capric acid. Stearic acid was the only FA, which was not detected by the bees (**Fig 4 and S2 Table**).

They could also differentiate between different individual FAs, i.e. omega-3 vs omega-6, omega-3 vs omega-9, omega-3 vs capric acid, omega-6 vs omega-9, omega-6 vs stearic acid, omega-6 vs capric acid, omega-9 vs stearic acid, omega-9 vs capric acid, and stearic acid vs capric acid (**Fig 5 and S2 Table**). The bees also showed a trend to differentiate between omega-3 and stearic acid, but this was not significant (**Fig 5 and S2 Table**). When capric acid was tested, the type of rewarded stimulus influenced the learning curves (**S3 Fig and S2 Table**).

Furthermore, the bees were able to perceive different concentrations of a FA mixture (1:100 FA mix and 1:1000 FA mix vs chloroform; **Fig 6 and S2 Table**). Interestingly, the lowest FA

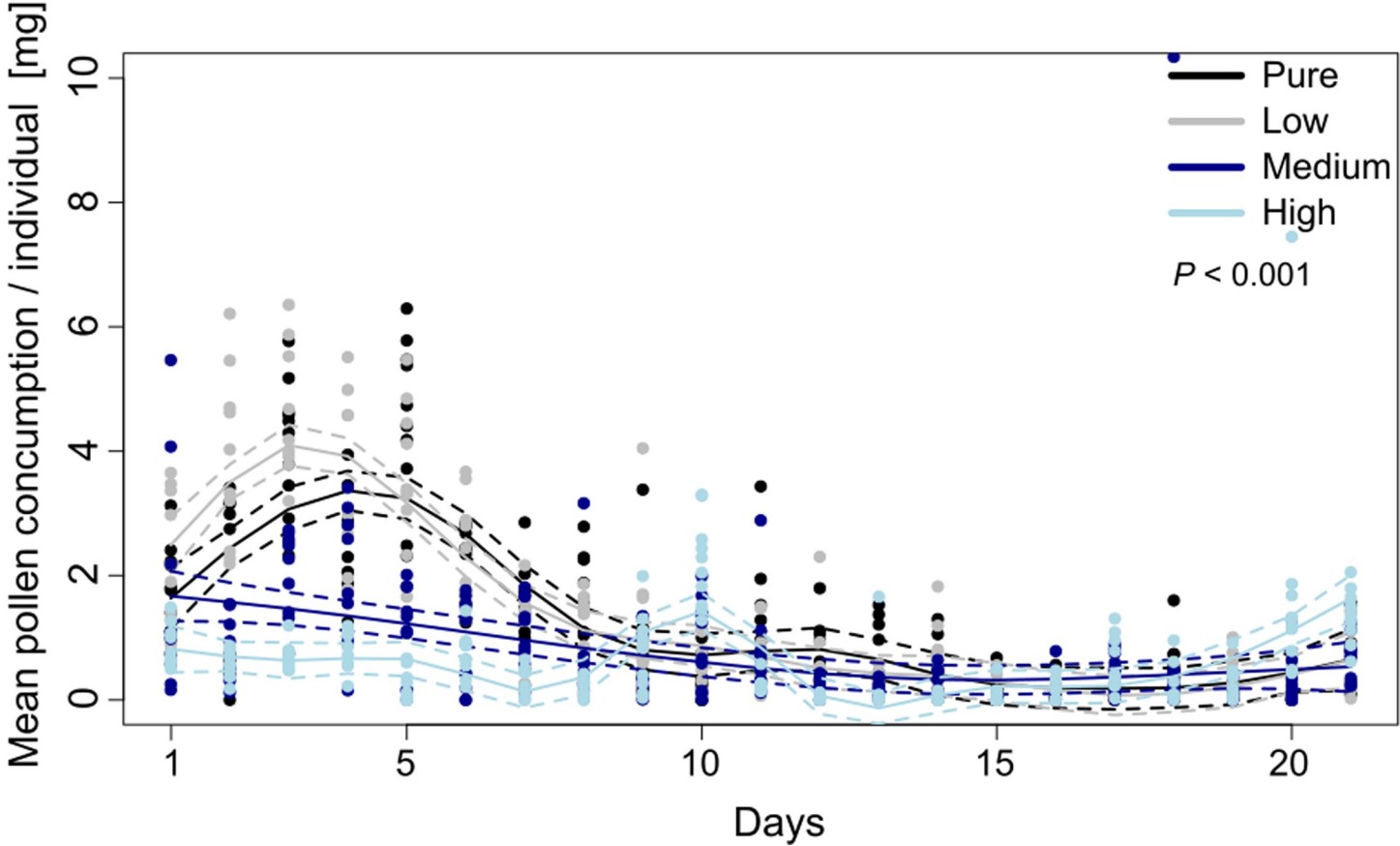

**Fig 1. Mean consumed food per individual and day of *Apis mellifera* workers fed with pollen diets differing in FA content (i.e. pure pollen, x1.5, x5 and x10 times the natural concentration) (N = 40).** Each dot represents the average pollen consumption per individual and day for one replicate. Continuous lines represent smoothers calculated by generalized additive models (GAMs) with dashed lines as upper and lower 95% confidence intervals. The different FA concentrations did affect the amount of food consumed per individual in ($F_{1,3}$ = 30.99, *P* = 0.001).

concentration (1:10000) was only perceived when the mixture was used as the rewarded stimulus (**Fig 6** and **S2 Table**). Finally, the bees could also discriminate between high and medium FA mix concentrations (**Fig 6** and **S2 Table**).

## Discussion

Our results show that *Apis mellifera* workers can perceive FAs and use this ability to regulate FA intake, i.e., they reduce the consumption of pollen enriched with medium to high concentrations of FAs. This led to a reduced survival probability of workers compared to workers feeding on low FA or control diets. Our results agree with findings of Ruedenauer [20], who showed similar effects in bumble bees offered pollen enriched with FA mixtures.

These findings indicate that high concentrations of FAs in pollen may be detrimental to young honey bees, explaining why their intake is strongly regulated by the bees, resulting in a reduced overall pollen consumption. This result adds to the increasing body of evidence that bees in general are adversely affected by an excess fat consumption at significantly lower levels as compared to proteins [47]. Elevated concentrations of FAs in food can restrict the absorption rate of FAs by midgut cells [47], potentially leading to the damage of cell membranes [48]. The composition and concentration of FAs consequently play an important role in bee health, which is analogous to human health, where saturated and trans-FAs increase the risk of

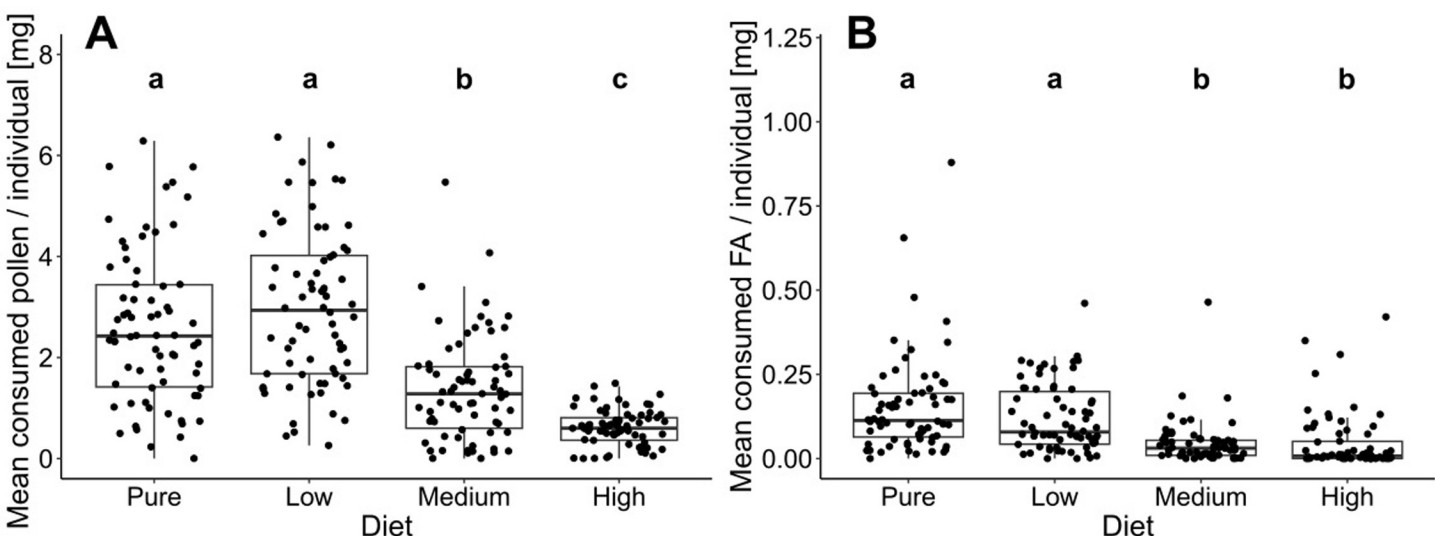

**Fig 2.** Mean consumption of (**A**) pollen and (**B**) FAs per individual and day during the first seven days after *Apis mellifera* worker bee emergence. Freshly emerged bees were fed with four different pollen diets differing in FA content (i.e. pure honey bee collected pollen as a control and the same pollen enriched with a FA-mix to achieve low FA pollen (1.5 times higher FA concentration), medium FA pollen (5 times higher FA concentration) and high FA pollen (10 times higher FA concentration)). Different letters above boxplots indicate significant differences in the mean consumption of pollen.

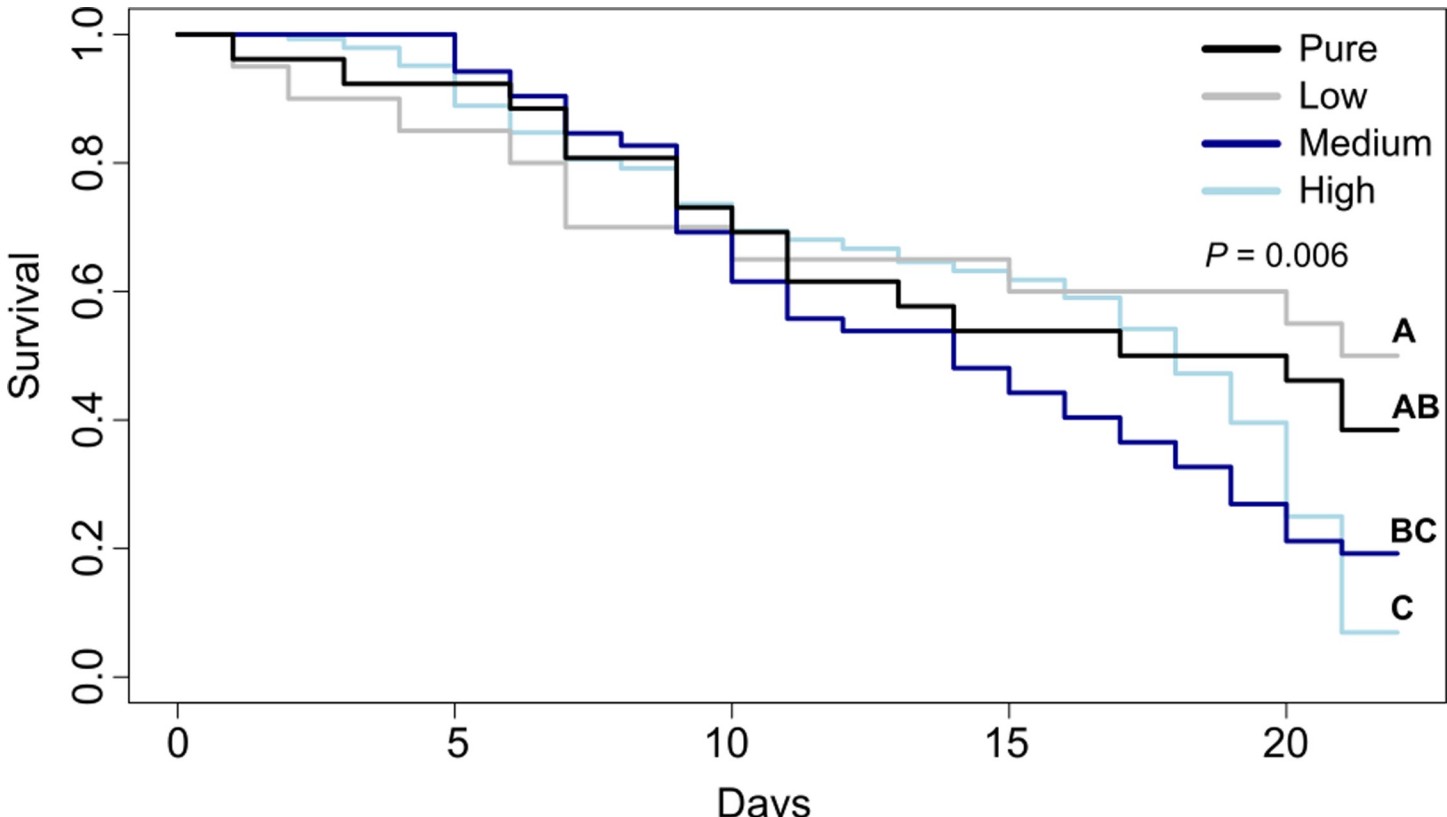

**Fig 3. Survival of freshly emerged *Apis mellifera* workers fed pollen diets differing in FA content (i.e., pure pollen or pollen enriched with x1.5, x5 or x10 the natural concentrations of a FA mixture) over a period of 21 days.** Different letters show significant differences regarding the survival probability for different diets (*P* = 0.006).

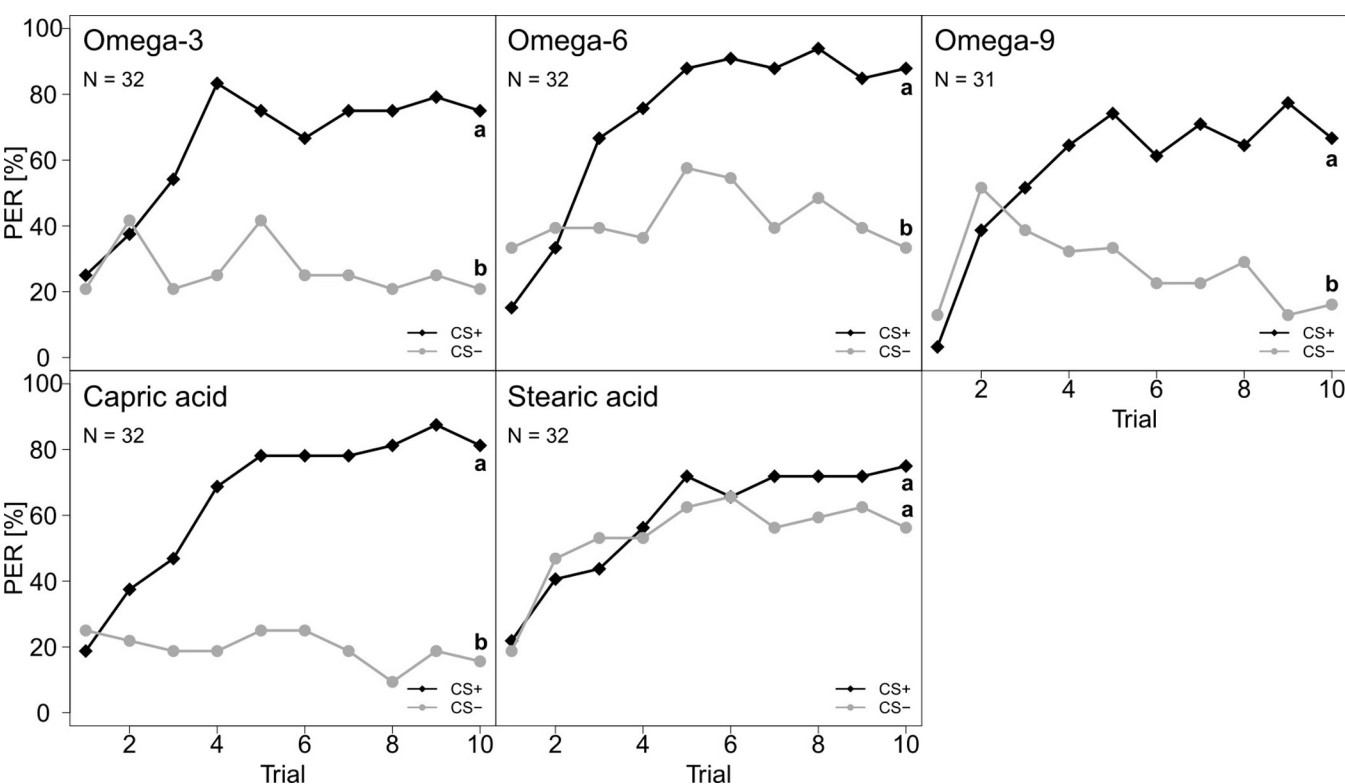

**Fig 4. Percentage of *A. mellifera* workers that responded with a proboscis extension response (PER) to one out of five FA when individually tested against the solvent control (chloroform).** From left to right and top to bottom: Omega-3, omega-6, omega-9, capric acid and stearic acid. CS+ (black, diamonds) represents the rewarded stimulus and CS- (grey, circles) the unrewarded stimulus. Both FA and solvent were tested as CS+ and CS-. Statistical differences are marked with different letters at the right side of the curves ($P < 0.05$).

coronary heart diseases, while mono- and polyunsaturated FAs (MUFAs and PUFAs) are linked with reduced risk [49, 50]. In comparatively low concentrations, low omega-6:3 ratios are also crucial for bee cognition [19, 23, 26]. In humans, the amount of omega-6 FAs consumed can modulate the amount of omega-3 FAs consumed and thereby reduce the amount of omega-3 available in the body [51], because excessive intake of omega-6 FAs interferes with the desaturation and elongation of omega-3 [52]. Humans therefore also favor an overall low omega-6:3 FA ratio.

The negative effects of high concentrations of specific FAs may explain why, in our study, bees exposed to medium and high FA diets drastically decreased their pollen intake and thereby consumed significantly less FAs compared to bees fed low FA or control diets. As a consequence, the reduced pollen intake may have led to an underconsumption of other essential nutrient compounds, like AAs, sterols, minerals and vitamins. Interestingly, bees seem to prioritize avoiding overconsumption of FAs even at the expense of potentially underconsuming other vital nutrients, which may explain why the medium and high FA groups showed a significant higher mortality compared to the control and low FA groups.

The presence and concentrations of different saturated FAs (SFAs) in pollen can strongly vary. For instance, myristic acid can constitute up to 47% of the total FA concentration in *Heliantheus annus* pollen [53]. Similarly, palmitic acid reaches up to 36% of the total pollen FA content in *Onobrychis viciifolia* pollen [53]. In contrast, SFAs, like lauric or stearic acid, are less commonly found but can still account for up to 13% of the total pollen FAs. Arachidic acid, on the other hand, is generally present in lower concentrations, with levels up to 5% of

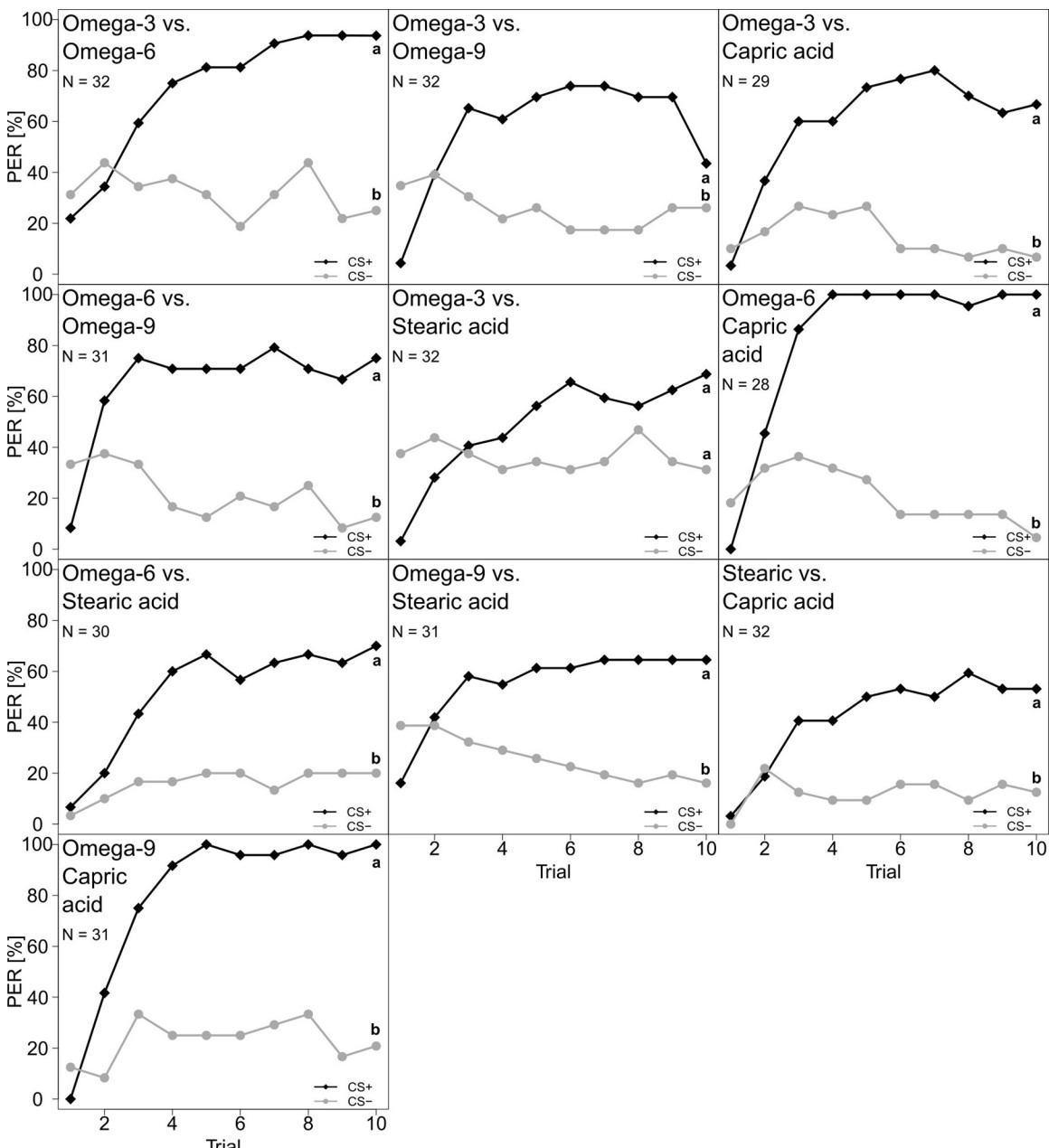

**Fig 5. Percentage of *A. mellifera* workers that responded with a PER to one out of ten FAs combinations.** From left to right and top to bottom: Omega-3 vs omega-6, omega-3 vs capric acid, omega-6 vs omega-9, omega-3 vs stearic acid, omega-6 vs capric acid, omega-6 vs stearic acid, omega-9 vs stearic acid, stearic acid vs capric acid and omega-9 vs. capric acid. CS+ (black, diamonds) represents the rewarded stimulus and CS- (grey, circles) the unrewarded stimulus. Both FA and solvent were tested as CS+ and CS-. Statistical differences are marked with different letters at the right side of the curves ($P < 0.05$).

the total FA content observed in the perennial monocot *Xanthorrhoea preissii* [53]. These concentrations align with concentrations found in the honey bee collected pollen used in our study. The concentrations used in our feeding experiments consequently were within the naturally occurring range observed in pollen, with the exception of the 10-fold higher concentration that exceeded natural levels.

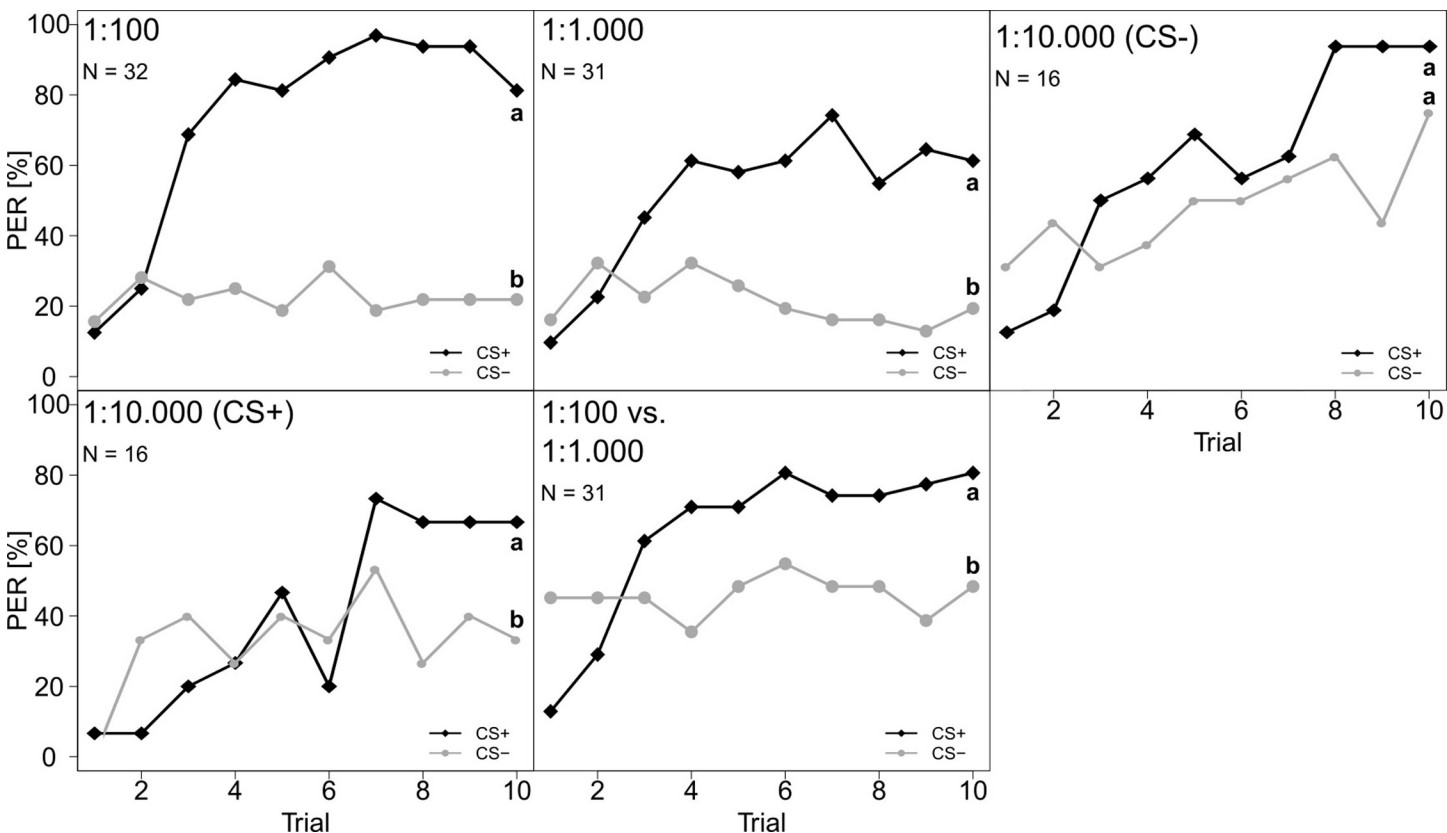

**Fig 6. Percentage of *A. mellifera* workers that responded with a PER to one out of four different concentrations of aFAs mixture against the solvent control chloroform.** From left to right and top to bottom: 1:100 FA mix vs chloroform, 1:1000 FA mix vs chloroform, 1:10000 FA mix as CS+ vs chloroform as CS- and 1:10000 FA mix as CS- vs chloroform as CS+. The last panel shows 1:100 FA mix vs 1:1000 FA mix. CS+ (black, diamonds) represents the rewarded stimulus and CS- (grey, circles) the unrewarded stimulus. Both FA and solvent were tested as CS+ and CS-. Statistical differences are marked with different letters at the right side of the curves (*P* < 0.05).

The varying importance of different FAs may further explain why honey bees were able to perceive and differentiate between FAs and different concentrations of the same FAs both in isolation and when added to pollen [28]. Moreover, honey bees could also discriminate among some PUFAs (linolenic, linoleic and oleic acid) which only differed in the number of double bounds but not in chain length. This precise discrimination implies that honey bees invest considerable resources into the reception and perception of FAs and subsequent behavioral responses. Across insects, perception of FAs relies on ionotropic (IR) and gustatory receptors (GR) [54–56]. In *Drosophila*, for example, the IR56d contributes to the perception of short-, medium-, and long-chained FAs [54, 55], while discrimination among chain lengths probably involves one or more receptor classes, such as IR56d+ and GR64f [55]. However, the receptors involved in FA reception in bees are completely unknown to date.

Notably, no such effect on intake and perception was observed when bumble bees were fed with pollen enriched with mixtures of AAs [20, 21] or sterols [31]. Variations in the concentrations of these nutrients did not influence reproductive fitness (i.e., number of produced larvae), even when consumed in concentrations beyond those naturally occurring in pollen [20, 31]. This supports the hypothesis that only those nutrients or compound groups, that are toxic or detrimental when over- or underconsumed, are perceived by bees prior to consumption. Pollen concentrations of other compounds which are essential for bee health and larval growth [57, 58], like AAs or sterols, are likely within the optimal or tolerated range and do therefore not

require specific attention by bees in terms of health effects [20, 27, 28, 31]. Interestingly, like bumble bees, honey bees did not differentiate between pollen diets enriched with different concentrations of sterols [28, 31]. However, unlike bumble bees, they discriminated between pollen diets enriched with different concentrations of AAs [28]. This difference between bumble bees and honey bees may suggest potential variations in nutrient-specific perception and likely even the importance of different nutrients for fitness across bee species. In fact, there is emerging evidence that taste receptor gene expression (for GR and IR) is even species-specific in bumble bees [59], which may reflect distinct nutritional requirements among bee species [60].

Overall, honey bees learned and discriminated different FAs equally well, except for stearic acid. Stearic acid, a saturated FA with a hydrocarbon chain length of 18 carbon atoms (same length as linoleic, linolenic and oleic acid), is one of the most common dietary FAs [61] including those found in pollen [62]. Interestingly, bumble bees were able to perceive stearic acid in isolation [20], but failed to regulate its intake as shown by no-choice feeding experiments, where diets were enriched with only one specific FA [63]. Compared to linoleic acid, stearic acid also scored lower intensity responses in humans when administered via edible taste stripes [64]. In fact, in mammals, stearic acid acts differently, i.e., generates a lower lipemic response (i.e., rise in triglycerides with lipoproteins following consumption) as compared to other shorter saturated FAs, like myristic or palmitic acid [65], which may be explained by the possible desaturation of stearic acid into the omega-9 MUFA oleic acid [65].

Honey bees were able to perceive and differentiate between capric acid both in isolation and when tested against other FAs. However, bees always learned to differentiate between the rewarded and unrewarded stimulus faster when capric acid was used as rewarded stimulus (see **S3 Fig** for an example of the differences in the learning curves). This explains why we had to always test the two complementary series involving capric acid (as CS+ and CS-) separately (**S2 Table**), and indicates that capric acid influenced the learning process. Capric acid is a medium-chain FA (ten carbon atoms), which can be rapidly absorbed and metabolized, likely rendering it a readily available and thus valuable energy source preferred by bees [66].

While our findings provide valuable insights into FA regulation by freshly emerged honey bees, we acknowledge the limitation of using only one colony in the feeding experiments, which may limit the generalizability of our results. However, our findings align with previous experiments conducted with bumble bees [20], supporting the validity of our data.

## Conclusion

Our results demonstrate the capacity of *A. mellifera* workers to effectively detect and discriminate between both saturated and unsaturated FAs by means of their antennae. They used this ability to regulate the intake of FAs, and when fed with high FA diets, they showed increased mortality rates due to their reduction of pollen and thus FA intake. Apparently, bees are willing to tolerate insufficient amounts of other nutrients to prevent overconsumption of FAs. Our findings add to the growing evidence of the importance of nutrition in maintaining bee health, fitness and survival. They also highlight the importance of a constantly provided diverse flowering plant spectrum, which bees can easily navigate through aided by their nutrient-sensitive perception prior to consumption. This enables them to adjust and optimize the nutrient composition of their diet according to their specific requirements [23, 67, 68]. A pre-ingestional perception of nutrients may thus represent a vital strategy employed by bees and likely other pollinators to strike a balance between overconsuming certain nutrients and underconsuming others in an environment full of nutritionally imbalanced food sources.

## Supporting information

**S1 Text. FA analysis.** Analysis of FA contents of honey bee collected pollen using gas chromatography- mass spectrometry.
(DOCX)

**S1 Table. Differences in pollen consumption and FA consumption.** Results of TukeyHSD post-hoc test analyzing differences in pollen (white background) and FA consumption (grey background) between diets enriched with low, medium and high FA concentrations for the first seven days.
(DOCX)

**S2 Table. Results of the chemotactile PER conditioning experiments.** All FAs were solved 1:1000 in chloroform. Each stimulus was tested as CS+ as well as CS- to test whether the rewarded stimulus influenced learning performance. If this was the case, data was not pooled but analyzed separately, otherwise it was pooled. To test if the bees can differentiate between two stimuli a paired Wilcoxon signed-rank test with continuity correction was used. N represents the number of bees tested per stimulus pair.
(DOCX)

**S1 Fig. Pollen and FA consumption during the second week.** Mean consumption of **(A)** pollen and **(B)** FAs per individual and day during day seven to 14. Bees were fed with four different pollen diets differing in FA content (i.e. pure honey bee collected pollen as a control and the same pollen enriched with a FA-mix to achieve low FA pollen (1.5 times higher FA concentration), medium FA pollen (5 times higher FA concentration) and high FA pollen (10 times higher FA concentration)). Different letters above boxplots indicate significant differences in the mean consumption of pollen.
(TIF)

**S2 Fig. Pollen and FA consumption during the third week.** Mean consumption of **(A)** pollen and **(B)** FAs per individual and day during day 15 to 21. Bees were fed with four different pollen diets differing in FA content (i.e. pure honey bee collected pollen as a control and the same pollen enriched with a FA-mix to achieve low FA pollen (1.5 times higher FA concentration), medium FA pollen (5 times higher FA concentration) and high FA pollen (10 times higher FA concentration)). Different letters above boxplots indicate significant differences in the mean consumption of pollen.
(TIF)

**S3 Fig. Percentage of *A. mellifera* workers that responded with a PER when capric acid was tested against omega-3. (A)** Omega-3 was used as the rewarded stimulus and capric acid as the unrewarded stimulus. **(B)** Capric acid was used as rewarded stimulus and omega-3 as unrewarded stimulus. CS+ (black, squares) represents the rewarded stimulus and CS- (grey, circles) the unrewarded stimulus. N represents the number of individuals tested. Statistical differences are marked with different letters at the right side of the curves ($P < 0.05$).
(TIF)

## Acknowledgments

We thank the entire TUM PII team and two anonymous reviewers for valuable feedback on a previous version of this manuscript.

## Author Contributions

**Conceptualization:** Fabian A. Ruedenauer, Sara D. Leonhardt, Johannes Spaethe.

**Data curation:** Marielle C. Schleifer, Johanna Ziegler.

**Formal analysis:** Marielle C. Schleifer.

**Funding acquisition:** Sara D. Leonhardt, Johannes Spaethe.

**Investigation:** Marielle C. Schleifer, Johannes Spaethe.

**Methodology:** Fabian A. Ruedenauer, Sara D. Leonhardt, Johannes Spaethe.

**Project administration:** Sara D. Leonhardt, Johannes Spaethe.

**Resources:** Sara D. Leonhardt, Johannes Spaethe.

**Supervision:** Fabian A. Ruedenauer, Sara D. Leonhardt, Johannes Spaethe.

**Validation:** Sara D. Leonhardt, Johannes Spaethe.

**Writing – original draft:** Marielle C. Schleifer.

**Writing – review & editing:** Fabian A. Ruedenauer, Johanna Ziegler, Sara D. Leonhardt, Johannes Spaethe.

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
