## [Decision Letter · Decision Letter 0]

10 Jun 2024

PONE-D-24-20571Perception, regulation, and fitness effects of pollen fatty acids in the honey bee, Apis melliferaPLOS ONE

Dear Dr. Leonhardt,

Thank you for submitting your manuscript to PLOS ONE. After careful consideration, we feel that it has merit but does not fully meet PLOS ONE’s publication criteria as it currently stands. Therefore, we invite you to submit a revised version of the manuscript that addresses the points raised during the review process.

We look forward to receiving your revised manuscript.

Kind regards,

Olav Rueppell

Academic Editor

PLOS ONE

Journal Requirements:

"Deutsche Forschungsgemeinschaft (DFG project: LE 2750/5-2 to SDL and SP1380/1-2 to JS)"

3. Please note that your Data Availability Statement is currently missing the repository name. If your manuscript is accepted for publication, you will be asked to provide these details on a very short timeline. We therefore suggest that you provide this information now, though we will not hold up the peer review process if you are unable.

**Additional Editor Comments:**

Most of the reviewers' concerns seem to stem from the need for more explanation/information (the most severe probably being the missing methods on FA determination) and a lack of attention to detail when preparing the manuscript (e.g., formatting). However, there are also a few comments that question your logic and I would encourage you to address everything very carefully because I will have to send the revised manuscript back to the reviewers for their input.

Reviewers' comments:

Reviewer's Responses to Questions

**Comments to the Author**

1. Is the manuscript technically sound, and do the data support the conclusions?

Reviewer #1: Partly

Reviewer #2: Yes

2. Has the statistical analysis been performed appropriately and rigorously? 

Reviewer #1: Yes

Reviewer #2: Yes

3. Have the authors made all data underlying the findings in their manuscript fully available?

Reviewer #1: No

Reviewer #2: Yes

4. Is the manuscript presented in an intelligible fashion and written in standard English?

Reviewer #1: Yes

Reviewer #2: Yes

5. Review Comments to the Author

Reviewer #1: The manuscript by Schleifer et al. examines the ability of honey bees to detect the various fatty acids that are present in pollen, their ability to regulate this and the potential fitness effects. The manuscript is very interesting and adds to the existing literature of how honey bees are able to regulate nutrient intake, including fatty acids. My general comments are:

1. Please mention the full form before the use of abbreviations throughout the manuscript.

2. Replace animals with colonies in L114.

3. L120 why not use multiple colonies and multiple frames to avoid colony specific behavior and physiological results?

4. The total experimental duration for the feeding experiment should be included in the results. I found this eventually when reading the figure legends.

5. How was FA concentration measured in pollen and throughout the experiment? A whole section is missing on how the authors evaluated the FA concentrations. Was mass spec used and where is the raw data?

6. L146 is there a physiological difference between honey bee foragers and their PER due to the long experimental duration and changing months?

7. What about the ages of the foragers? How was forager age controlled for?

8. More clarity in the method section is needed for better understanding the methods and for the benefit of the readers. L 148 were sister queens used? What was the total number of hives eventually used? Were the same newly emerged honey bees tested for all the experimental replicates? How many were tested for each group?

9. Similar to the experimental duration, methods do not indicate that chloroform was tested as a solvent control for PER. I found this later on in the manuscript. Please include this to help improve the method section for further clarity.

10. L230 for the benefit of the readers the authors should indicate how they calculated FA consumption data.

11. Please add the weekly data for the entire experimental duration even though the first week showed most consumptions. It will be interesting to see what happens every week. This will also add more information on how much pollen was consumed for each experimental group. And how much FA was also consumed.

12. L252 please italicize scientific names.

13. L254 the graph shown does not seem to match 21 days for experimental duration based on the axis.

14. For the PER experiments in general, are honey bees not trained with a sugar reward as a positive stimulus response and no sugar reward as a negative response? So it is expected that honey bees will learn and not respond when sugar reward is not presented? Even if FA were presented? Also was the same honey bee tested for all experimental groups in all combinations (solvent control and each FA as CS+ and CS-)? If not how were honey bees age controlled/colony controlled to negate any other bias? How many training attempts and how many test attempts for each honey bee?

15. All raw data have not been presented and please include that.

16. While survival is a measure of fitness, not sure if the authors can truly use “fitness” in their manuscript title without actually measuring the various indices of fitness for honey bees (physiology and colony health).

Reviewer #2: The authors should make a few changes before the document is forwarded to reviewers.

- please check the reference format in the overall manuscript.

- I think chloroform is toxic to honey bees. Did you check the safety of this chemical for honey bees? We attached the PDF which is chloroform MSDS

-According to the PER experiment, except for the combination of omega-3 and stearic acid, honeybees were able to distinguish each fatty acid. However, concluding that honeybees distinguish between saturated and unsaturated fatty acids seems unreasonable.

- Line 105: Please change ‘overconsumption’ to ‘High amount of FA content’.

- Line 128: "0.5 times" means a decrease, not an increase. Therefore, please change all instances of 0.5 to 1.5.

- Line 132: Please explain more clearly about the FA mixture, for example, where is the source (which company?) or Did you mix yourself using each FAs then, please list the company of each FA, for example: palmitic (Sigma-Aldrich, St. Louis, MO).

- Line 142: In Table 1, How did you measure the fatty acids content in pollen? And what is FS? Maybe FS should be changed to FA.

- Line 237: Please add statistical results in Figure 2 (A and B). Please add the statistical result in the manuscript even present in the S1 Table.

- Line 252: Please check that 'Apis mellifera' is in italics in all manuscripts.

- Line 264: In Figure 3, please explain why CS- in stearic acid showed higher than other FAs.

- Line 268: Why did you choose to PER test linoleic, linolenic, oleic, capric, and stearic acids among the fatty acids?

- Line 271: Please check that 'P' is in italics in all manuscripts.

- Line 278: In Figure 4, Please list the full name of FAs, for example, ‘Omega-6 Capric’ change to ‘Omege-6 vs. Capric acid’. Please upgrade Figure 4.

- Line 291: Please change ‘1:1000 vs chloroform’ to ‘1:1000 FA mix vs chloroform’.

- In the discussion, please add more explanation of the percentage of each fatty acid (except Omega-3, Omega-6, and Omega-9) included in general pollen and explain the effect on honeybees.

- Line 300: Please delete the second author name.

- Line 305, 323, 332: Please delete the author's name.

6. PLOS authors have the option to publish the peer review history of their article (what does this mean?). If published, this will include your full peer review and any attached files.

Reviewer #1: No

Reviewer #2: No

---

## [Author Response · Author response to Decision Letter 0]

20 Jul 2024

Thank you very much for the possibility to submit a revised version of our manuscript to PLOS ONE. We also thank the two reviewers for their valuable and constructive comments on the manuscript. We have addressed all points raised by the reviewers and provide a detailed response to each comment individually below (starting with --).

Reviewer #1: The manuscript by Schleifer et al. examines the ability of honey bees to detect the various fatty acids that are present in pollen, their ability to regulate this and the potential fitness effects. The manuscript is very interesting and adds to the existing literature of how honey bees are able to regulate nutrient intake, including fatty acids. 

-- Thank you very much for this remark, and also for taking the time to review our manuscript and for providing constructive feedback.

My general comments are:

1. Please mention the full form before the use of abbreviations throughout the manuscript.

-- Done.

2. Replace animals with colonies in L114.

-- Done.

3. L120 why not use multiple colonies and multiple frames to avoid colony specific behavior and physiological results?

-- Thank you for pointing that out, and we agree that this would be the optimal procedure. Unfortunately, at the time period of the experiment, we had only one colony with one mature brood frame available. Since this single brood frame had a sufficient number of bees emerged on the same day, we decided to stick with only one brood frame from a single colony. We now clarified this aspect in the manuscript and discuss the limitations of using only one colony.

4. The total experimental duration for the feeding experiment should be included in the results. I found this eventually when reading the figure legends.

-- The experimental duration is mentioned in the first sentence of the ‘Feeding experiment’ section (L127), but for clarification, we now also added the duration in the ‘Results’ section.

5. How was FA concentration measured in pollen and throughout the experiment? A whole section is missing on how the authors evaluated the FA concentrations. Was mass spec used and where is the raw data?

-- Thank you for pointing this out! We added the information of the FA analysis to the ‘Material & Method’ section and included a detailed protocol in the Supplementary Material section!

6. L146 is there a physiological difference between honey bee foragers and their PER due to the long experimental duration and changing months?

-- We could see differences in the response of foragers during the season. For example, early in the season, when many flowering plants were available, less bees were willing to show a PER when touching the antenna during the pre-test, compared to later in the season. However, only bees that showed a proper PER and thus a high motivational state, were used in the experiments. We additionally tested all stimuli randomly throughout the season to avoid any bias.

7. What about the ages of the foragers? How was forager age controlled for?

-- We randomly caught the bees while leaving the hive, which is why we could not control for age. However, studies showed, that honey bees’ cognitive abilities are influenced by their social role rather than the chronological age. We have added this information in the manuscript.

8. More clarity in the method section is needed for better understanding the methods and for the benefit of the readers. L 148 were sister queens used? What was the total number of hives eventually used? Were the same newly emerged honey bees tested for all the experimental replicates? How many were tested for each group?

-- In total, we used bees from ten hives (L158), but no sister queens were used. 

Newly emerged bees were only used for the feeding experiment since the consumption of pollen by adult bees declines rapidly after the first week. Each bee was assigned to only one dietary treatment. We had 4 treatments with 10 replicates each. Each replicate comprised 30 newly emerged bees – so 300 bees for each dietary treatment (L131-133). Bees used in the feeding experiments were not used for the learning / PER experiments; only foragers that were about to leave the hive, were used for PER experiments. All this information has now been added/clarified in the methods.

9. Similar to the experimental duration, methods do not indicate that chloroform was tested as a solvent control for PER. I found this later on in the manuscript. Please include this to help improve the method section for further clarity.

-- Thank you for the hint! We now added this information to the ’Stimuli’ paragraph of the ‘Material and Method’ section.

10. L230 for the benefit of the readers the authors should indicate how they calculated FA consumption data.

-- Good point. This is now included in the beginning of the ‘Statistical analysis’ section.

11. Please add the weekly data for the entire experimental duration even though the first week showed most consumptions. It will be interesting to see what happens every week. This will also add more information on how much pollen was consumed for each experimental group. And how much FA was also consumed.

-- We agree and have now added the respective figures in the Supplementary Material

12. L252 please italicize scientific names.

-- Done.

13. L254 the graph shown does not seem to match 21 days for experimental duration based on the axis.

-- Thank you for the hint. We have revised the figure. 

14. For the PER experiments in general, are honey bees not trained with a sugar reward as a positive stimulus response and no sugar reward as a negative response? So it is expected that honey bees will learn and not respond when sugar reward is not presented? Even if FA were presented? Also was the same honey bee tested for all experimental groups in all combinations (solvent control and each FA as CS+ and CS-)? If not how were honey bees age controlled/colony controlled to negate any other bias? How many training attempts and how many test attempts for each honey bee?

-- You are correct, the rewarded stimulus (CS+) is combined with sugar water (e.g. fatty acid A + sugar water), and the unrewarded stimulus (CS-) is not paired with anything but the stimulus alone (e.g. fatty acid B). 

Naïve, untrained bees will only elicit a PER towards sugar water and not towards a FA alone. In a classical absolute conditioning experiment, when the FA is paired with the sugar water, the bee associates the sugar reward with this FA after repeated presentations and will elicit a PER to the FA alone, but only if the bee can perceive the FA. In a classical differential conditioning experiment, one FA will be rewarded (CS+), a second FA will be unrewarded (CS-). After repeated random presentations of the CS+ and CS-, the bee will elicit a PER only to the rewarded one, but no PER to the unrewarded one.

Regarding testing: an individual bee was tested only once (described in L184), otherwise the bee would already have learned one stimulus and cannot be considered a naïve, untrained bee. 

We randomly took bees from different colonies to exclude any colony bias, but we did not check for age (although leaving forager bees were considered to be between 16 and 35 day old). Each bee was tested ten times for the rewarding and ten times for the unrewarded stimulus, in total 20 trials (L 196-197).

We have tried to clarify these details throughout the manuscript.

15. All raw data have not been presented and please include that.

-- The raw data for the FA analysis of the pollen has now been added and is available on https://doi.org/10.17605/osf.io/geuzv.

16. While survival is a measure of fitness, not sure if the authors can truly use “fitness” in their manuscript title without actually measuring the various indices of fitness for honey bees (physiology and colony health).

-- We agree with the reviewer and have changed the title to: Perception, regulation, and fitness effects on longevity of pollen fatty acids in the honey bee, Apis mellifera

Reviewer #2: The authors should make a few changes before the document is forwarded to reviewers.

1. please check the reference format in the overall manuscript.

-- Thank you for this hint; we checked for all mistakes and have now a consistent reference format.

2. I think chloroform is toxic to honey bees. Did you check the safety of this chemical for honey bees? We attached the PDF which is chloroform MSDS

-- This is correct; however, we have made sure that the chloroform evaporates completely before the filter paper was presented to the bees. We now added this important information to the Method section. 

3. According to the PER experiment, except for the combination of omega-3 and stearic acid, honeybees were able to distinguish each fatty acid. However, concluding that honeybees distinguish between saturated and unsaturated fatty acids seems unreasonable.

-- We agree that our conclusion may have been too far-fetched and have adjusted our wording in the revised version (L358-359).

4. Line 105: Please change ‘overconsumption’ to ‘High amount of FA content’.

-- Done.

5. Line 128: "0.5 times" means a decrease, not an increase. Therefore, please change all instances of 0.5 to 1.5.

-- Done.

6. Line 132: Please explain more clearly about the FA mixture, for example, where is the source (which company?) or Did you mix yourself using each FAs then, please list the company of each FA, for example: palmitic (Sigma-Aldrich, St. Louis, MO).

-- Thank you for the hint. We added the source of the single FAs.

7. Line 142: In Table 1, How did you measure the fatty acids content in pollen? And what is FS? Maybe FS should be changed to FA.

-- Thank you for pointing this out. Of course, FS should be stated as FA, which we have changed now. We now also mention the FA analysis in the Material and Methods section and added the full protocol in the Supplementary Material.

8. Line 237: Please add statistical results in Figure 2 (A and B). Please add the statistical result in the manuscript even present in the S1 Table.

-- Done.

9. Line 252: Please check that 'Apis mellifera' is in italics in all manuscripts.

-- Done.

10. Line 264: In Figure 3, please explain why CS- in stearic acid showed higher than other FAs.

-- We do not know why the bees did not perceive stearic acid, which we mention and discuss in the text. 

11. Line 268: Why did you choose to PER test linoleic, linolenic, oleic, capric, and stearic acids among the fatty acids?

-- We added our rationale to the Material and Method section, which now reads: “We selected these five FA to test if the bees can differentiate between FAs based on the chain length, saturation state or the number of double bounds”.

12. Line 271: Please check that 'P' is in italics in all manuscripts.

-- Done.

13. Line 278: In Figure 4, Please list the full name of FAs, for example, ‘Omega-6 Capric’ change to ‘Omege-6 vs. Capric acid’. Please upgrade Figure 4.

-- Done.

14. Line 291: Please change ‘1:1000 vs chloroform’ to ‘1:1000 FA mix vs chloroform’.

-- Done.

15. In the discussion, please add more explanation of the percentage of each fatty acid (except Omega-3, Omega-6, and Omega-9) included in general pollen and explain the effect on honeybees.

-- Thank you for mentioning it. We have added this information in the discussion 

16. Line 300: Please delete the second author name.

-- Done.

17. Line 305, 323, 332: Please delete the author's name.

-- Done.

---

## [Decision Letter · Decision Letter 1]

12 Aug 2024

PONE-D-24-20571R1Perception, regulation, and effects on longevity of pollen fatty acids in the honey bee, Apis melliferaPLOS ONE

Dear Dr. Leonhardt,

Thank you for re-submitting your manuscript to PLOS ONE. As you can see, the reviewer (and I) are satisfied with your revisions. However, before I can fully accept it, please fix the references at this stage. You might also want to give the manuscript an overall read through for catching minor errors. Finally, if you could please explain the authorship change?

We look forward to receiving your revised manuscript.

Kind regards,

Olav Rueppell

Academic Editor

PLOS ONE

Journal Requirements:

Reviewers' comments:

Reviewer's Responses to Questions

**Comments to the Author**

1. If the authors have adequately addressed your comments raised in a previous round of review and you feel that this manuscript is now acceptable for publication, you may indicate that here to bypass the “Comments to the Author” section, enter your conflict of interest statement in the “Confidential to Editor” section, and submit your "Accept" recommendation.

Reviewer #2: All comments have been addressed

2. Is the manuscript technically sound, and do the data support the conclusions?

Reviewer #2: Yes

3. Has the statistical analysis been performed appropriately and rigorously? 

Reviewer #2: Yes

4. Have the authors made all data underlying the findings in their manuscript fully available?

Reviewer #2: Yes

5. Is the manuscript presented in an intelligible fashion and written in standard English?

Reviewer #2: Yes

6. Review Comments to the Author

Reviewer #2: Revision has been made according to reviewers' comments, but in information on references 13, 29, 30, no page numbers are found, please add it.

7. PLOS authors have the option to publish the peer review history of their article (what does this mean?). If published, this will include your full peer review and any attached files.

Reviewer #2: **Yes: **Hyung Wook Kwon

---

## [Author Response · Author response to Decision Letter 1]

18 Aug 2024

Dear Hyung Wook Kwon,

thank you very much for taking them time and effort to once again read our manuscript and for your valuable feedback on the manuscript. 

We have changed the reference list and added the missing information.

Warmest regards,

Sara Leonhardt and all co-authors

---

## [Editor Report · Decision Letter 2]

20 Aug 2024

Perception, regulation, and effects on longevity of pollen fatty acids in the honey bee, Apis mellifera

PONE-D-24-20571R2

Dear Dr. Leonhardt,

We’re pleased to inform you that your manuscript has been judged scientifically suitable for publication and will be formally accepted for publication once it meets all outstanding technical requirements.

Kind regards,

Olav Rueppell

Academic Editor

PLOS ONE
---

## [Editor Report · Acceptance letter]

27 Aug 2024

PONE-D-24-20571R2 

PLOS ONE

Dear Dr. Leonhardt, 

I'm pleased to inform you that your manuscript has been deemed suitable for publication in PLOS ONE. Congratulations! Your manuscript is now being handed over to our production team.

Kind regards, 

on behalf of

Dr. Olav Rueppell 

Academic Editor

PLOS ONE